# Consumer understanding of terms used in imaging reports requested for low back pain: a cross-sectional survey

Caitlin Farmer [1,2] Denise A O'Connor [1,2] Hopin Lee [3,4]
Kirsten McCaffery [5] Christopher Maher [6,7] Dave Newell [8]
Aidan Cashin [9,10] David Byfield [11] Jeffrey Jarvik [12,13]
Rachelle Buchbinder [1,2]

For numbered affiliations see end of article.

**Correspondence to**
Caitlin Farmer;
caitlin.farmer1@monash.edu

## ABSTRACT

**Objectives** To investigate (1) self-reported societal comprehension of common and usually non-serious terms found in lumbar spine imaging reports and (2) its relationship to perceived seriousness, likely persistence of low back pain (LBP), fear of movement, back beliefs and history and intensity of LBP.

**Design** Cross-sectional online survey of the general public.

**Setting** Five English-speaking countries: UK, USA, Canada, New Zealand and Australia.

**Participants** Adults (age >18 years) with or without a history of LBP recruited in April 2019 with quotas for country, age and gender.

**Primary and secondary outcome measures** Self-reported understanding of 14 terms (annular fissure, disc bulge, disc degeneration, disc extrusion, disc height loss, disc protrusion, disc signal loss, facet joint degeneration, high intensity zone, mild canal stenosis, Modic changes, nerve root contact, spondylolisthesis and spondylosis) commonly found in lumbar spine imaging reports. For each term, we also elicited worry about its seriousness, and whether its presence would indicate pain persistence and prompt fear of movement.

**Results** From 774 responses, we included 677 (87.5%) with complete and valid responses. 577 (85%) participants had a current or past history of LBP of whom 251 (44%) had received lumbar spine imaging. Self-reported understanding of all terms was poor. At best, 235 (35%) reported understanding the term 'disc degeneration', while only 71 (10.5%) reported understanding the term 'Modic changes'. For all terms, a moderate to large proportion of participants (range 59%–71%), considered they indicated a serious back problem, that pain might persist (range 52%–71%) and they would be fearful of movement (range 42%–57%).

**Conclusion** Common and usually non-serious terms in lumbar spine imaging reports are poorly understood by the general population and may contribute to the burden of LBP.

**Trial registration number** ACTRN12619000545167.

### Strengths and limitations of this study

► This is the first study to directly measure societal self-reported understanding of 14 commonly used terms found in lumbar spine imaging reports and its relationship to worry regarding the seriousness and persistence of low back pain and being fearful of movement.

► Participants were a large representative sample from five high-income English-speaking countries with quotas for age and gender suggesting our results are likely generalisable to those settings but may or may not be generalisable to other settings.

► Self-reported understanding may not truly reflect a person's understanding of the terms, and the use of a survey sampling company may have introduced some bias.

► Although all terms that we investigated are findings that commonly exist in the asymptomatic population, some are more common in people with low back pain, and this may have influenced participants' responses.

global disability.[1] Most LBP is labelled 'non-specific', reflecting an inability to identify a specific cause in most cases.[2] In the absence of features suggestive of a serious cause, clinical practice guidelines consistently recommend against any form of lumbar spine imaging as it is unlikely to provide meaningful information about the cause of pain or guide treatment.[3–5] Imaging often reveals age-related changes such as disc degeneration, disc bulge and facet joint degeneration that are also common in asymptomatic people.[6]

Lumbar spine imaging also carries risk of iatrogenic harm. This includes not only exposure to ionising radiation with plain radiographs and CT scans, but the way findings are described may lead to clinician and patient anxiety about perceived seriousness, prompt further unnecessary tests, unwarranted diagnostic labels and unnecessary treatment.[7 8]

## INTRODUCTION

Low back pain (LBP) is a common, costly problem and a significant contributor to

While few published studies have investigated consumer understanding and views regarding terminology in lumbar spine imaging findings in people with LBP, several studies have investigated these issues for other conditions.[9–14] A systematic review investigating how different terms for various conditions influence management preferences found that use of more medicalised or precise terms generally results in greater anxiety and perceived severity, and preferences towards more invasive care.[15] Ambiguous terminology for incidental low-risk liver lesions reported in abdominal CT reports was found to increase patient and referrer anxiety and a greater likelihood of follow-up testing.[16] In contrast, simplified upper limb MRI reports using more understandable and less 'emotive' language (eg, 'expected age-related changes' rather than 'hypertrophic degenerative changes') have been found to reduce worry, improve comprehension and help patients feel more in control of their health.[17]

As patients are increasingly accessing their imaging reports, it is important to understand how commonly reported terms are interpreted by patients. The primary aim of this study was to investigate self-reported understanding of terms that are both commonly found in lumbar spine imaging reports and are usually of little clinical relevance. We also investigated the relationship between self-reported understanding of these terms and their perceived seriousness, perceptions about the likelihood of pain persistence and whether they would result in being fearful of movement. Our secondary aim was to describe the association between self-reported understanding of the terms and back beliefs, and history and intensity of LBP. We also elicited views about access to imaging reports and whether they should be understandable to patients and include context.

## METHODS
### Design
This study was a cross-sectional online survey of a random sample of the general public in five English-speaking countries: Australia, New Zealand, the USA, UK and Canada. It was conducted and reported in accordance with the Checklist for Reporting Results of Internet E-Surveys guideline.[18]

### Participants and recruitment
Participants were recruited in April 2019 via an online survey sampling company (Dynata). Dynata is a data company that maintains a research and marketing database. It recruits participants via email based on the requirements of the research involved. On survey completion, Dynata provides participant incentives that allow redemption of items such as gift cards and contributions to charity. English-speaking adults aged 18 and over with or without a current or past history of back pain were recruited via email with quotas established for country, age and gender to ensure a representative sample.

The survey was conducted in Qualtrics and no identifying information was provided to the research team.

### Survey instrument
Pilot testing conducted in a convenience sample of ten members of the general public resulted in minor changes. The survey included nine demographic questions (age in years, gender (female/male/other/prefer not to say), education level (finished high school/technical/trade certificate or diploma, university (bachelor), postgraduate university qualifications), employment status (full time/part time/unemployed/student/home duties/retired (not due to ill health)/retired (due to ill health)/other), healthcare provider for people with LBP (yes/no), LBP history (current, past, never) and for those with a history of LBP we asked about average pain intensity (measured on a 0–100 point Visual Analogue Scale (VAS) where higher scores indicate greater pain), LBP imaging (yes/no/don't know) and treatment (surgery and injections).

Four statements were presented for each of 14 radiology terms (annular fissure, disc bulge, disc degeneration, disc extrusion, disc height loss, disc protrusion, disc signal loss, facet joint degeneration, high intensity zone, mild canal stenosis, Modic changes, nerve root contact, spondylolisthesis and spondylosis). These terms were chosen on the basis that they are commonly found in lumbar imaging reports of people with LBP but are also common findings in asymptomatic populations and are usually considered benign or of unclear clinical significance.[6 19 20] For each question, order of presentation of the terms was randomised to reduce order effect bias.

For each item, we elicited the participant's level of agreement on a five-point Likert scale (1=completely disagree, 2=somewhat disagree, 3=neither agree nor disagree, 4=somewhat agree, 5=completely agree) to the following four statements:
1. I would need to look this term up to know what it means (Understanding).
2. I would be worried there is a serious problem with my back (Seriousness).
3. I would be worried my pain isn't going to get any better (Persistence).
4. I would be afraid to move my back in case I did more damage (Fear of movement).

Items 2 and 3 were derived from questionnaires for musculoskeletal conditions or LBP that assess perceived seriousness or likely persistence of pain.[21–23] For example, the third item is similar to items in the STarT Back Tool ('I feel my back pain is terrible and it's never going to get any better'),[24] Pain Catastrophising Scale ('It's terrible and I think it's never going to get any better')[25] and the Back Pain Attitude Questionaire (Back-PAQ) ('There is a high chance that an episode of back pain will not resolve').[22]

We also elicited opinions (yes/no/don't know) about whether people should have access to their imaging reports; whether imaging reports should be written so they can be understood by lay people; and whether

including epidemiological information about common findings in people without back pain would be helpful in interpreting results.

The Back Beliefs Questionnaire (BBQ) is a 14-item tool that measures beliefs about back pain.[26] Responses to each item are measured on a five-point Likert agreement scale (completely disagree to completely agree). The BBQ is scored from nine of the 14 items with higher scores indicating more positive beliefs (score range 9–45).

## Sample size and analysis

A minimum sample size of 590 was required to detect 70%±10% prevalence of not understanding a term. Assuming 10% drop-out or non-completion of surveys, our target was 650 participants. This sample size also provided enough power to investigate associations between variables.

Raw data files were reviewed for evidence of duplicate answers including duplicate internet protocol addresses prior to inclusion in the analysis. Descriptive statistics were used to summarise the participant characteristics, responses to statements about radiological terms, back beliefs and opinions regarding imaging reports. Data were excluded for any participant who provided the same answer for every item of the BBQ as this type of response is invalid and likely indicates that the questionnaire responses for other items were also invalid.

Responses of either completely disagree or somewhat disagree with the statement 'I would need to look this up to know what it means' were considered to indicate self-reported understanding of a term.

Responses for each item were dichotomised with 'completely disagree' and 'somewhat disagree' coded as zero and 'neither agree nor disagree'/'somewhat agree'/'completely agree' coded as one. For self-reported understanding, we combined a neutral response (neither agree nor disagree) with somewhat or completely agree that the person would need to look the term up to know what it means as we considered a neutral response more likely indicated that a respondent didn't understand a term. For items regarding worry about seriousness, persistence of pain and fear of movement, we combined a neutral response with somewhat or completely agree with these statements (ie, worried about seriousness, pain persistence or being fearful of movement).

The total number and per cent of positive responses for all 14 radiology terms were calculated for each participant. For the item about understanding, a score of 0 (0%) indicates that respondents reported that they would not need to look up any of the 14 terms (high level of understanding), while a score of 14 (100%) indicates they would need to look up all terms (low understanding). For items regarding worry about seriousness, persistence of pain and fear of movement, a score of 0 indicates that none of the terms elicited worry about seriousness, persistence of pain and/or movement, while a score of 14 indicates that all terms elicited concerns about seriousness, persistence of pain and/or movement.

Spearman's rank correlation was used to investigate the correlation between responses to the four statements regarding the terms. Multivariate regression analysis was used to investigate the association between self-reported understanding of terms and perceived seriousness, risk of persistence and fear of movement (measured by proportion of positive responses), and their association with participant demographics, history of LBP and BBQ score.

## Sensitivity analysis

To ensure that the dichotomisation was an appropriate method of handling the data, in particular the neutral responses, we performed two post hoc sensitivity analyses following comments from the reviewers of the manuscript. In the first, for the questions regarding seriousness, pain persistence and fear of movement, we combined the neutral responses with the strongly disagree and somewhat disagree responses and compared them to the somewhat agree and strongly agree responses. In the second sensitivity analysis, we removed neutral responses altogether and compared the answers of strongly disagree and somewhat disagree with somewhat agree and strongly agree.

## Patient and public involvement

Consumer feedback about questionnaire burden and time requirements was incorporated into survey design. Survey participants were advised how to access results of the study in the accompanying explanatory statement.

## RESULTS

Of 774 responses, we excluded 97 (12.5%) due to incomplete data (n=35, 4.5%), invalid BBQ responses (n=59, 7.6%) and age given as under 18 years (n=3, 0.4%). No duplicate responses were identified. We were unable to calculate a response rate as Dynata was unable to provide precise data about how often the survey was available to users via either email or their dashboard.

Responses from 677 participants (52% female) were analysed (table 1). There were no significant between-country differences regarding demographic characteristics. A total of 577 participants (85%) reported past or current LBP, and 279 (41%) reported having LBP at the time of survey completion. The mean (SD) BBQ score for the whole cohort was 25.0 (6.4). Pain intensity was higher in those who reported current compared with past LBP (mean (SD): 54.7 (21.3), n=279, vs 44.2 (22.1), n=298, respectively, p<0.001). Of those with a history of LBP, 251 (44%) reported receipt of lumbar spine imaging, 81 (14%) reported receipt of spine injection/s and 27 (5%) reported undergoing spine surgery.

## Participant views regarding terms

Most participants (n=598, 88%) somewhat or completely agreed they would need to look up at least one of the terms to know what it means, while only 3% (n=17) somewhat or completely disagreed with this statement for all

**Table 1** Participant demographics and back pain history and beliefs by country and overall

| | | USA (n=139) | Australia (n=138) | NZ (n=135) | UK (n=132) | Canada (n=133) | Total (n=677) |
|---|---|---|---|---|---|---|---|
| | | Mean (SD) | Mean (SD) | Mean (SD) | Mean (SD) | Mean (SD) | Mean (SD) |
| Age, years | | 45.4 (16.0) | 45.0 (17.0) | 45.9 (17.6) | 47.6 (17.0) | 45.4 (15.5) | 45.8 (16.4) |
| Back pain intensity most recent episode* (0–100 scale, higher score indicates greater pain) | Current LBP | 49.7 (22.4) | 58.2 (20.2) | 55.0 (21.7) | 58.1 (22.0) | 51.9 (20.2) | 54.7 (21.3) |
| | Previous LBP | 40.0 (19.7) | 48.4 (22.6) | 44.2 (21.3) | 45.4 (24.4) | 44.0 (22.6) | 44.2 (22.1) |
| BBQ score (9–45 scale, higher score indicates better back beliefs) | | 25.7 (6.4) | 24.3 (6.5) | 25.8 (6.6) | 24.6 (6.3) | 24.4 (6.1) | 25.0 (6.4) |
| | | N (%) | N (%) | N (%) | N (%) | N (%) | N (%) |
| Female | | 75 (54) | 73 (53) | 70 (52) | 67 (51) | 69 (52) | 354 (52) |
| University education | | 69 (50) | 51 (37) | 63 (47) | 59 (45) | 57 (43) | 299 (44) |
| Employed full or part time | | 71 (51) | 80 (58) | 77 (57) | 73 (55) | 80 (60) | 381 (56) |
| Worked as healthcare professional† | | 10 (7) | 4 (3) | 4 (3) | 5 (4) | 4 (3) | 27 (4) |
| Ever had low back pain | | 116 (83) | 119 (86) | 114 (84) | 113 (86) | 115 (86) | 577 (85) |
| Current low back pain | | 54 (38) | 68 (49) | 45 (33) | 52 (39) | 60 (45) | 279 (41) |
| Had lumbar spine imaging* | | 60 (52) | 60 (50) | 40 (35) | 38 (34) | 53 (46) | 251 (44) |
| Had lumbar spine injection* | | 17 (15) | 18 (15) | 16 (14) | 21 (19) | 9 (8) | 81 (14) |
| Had lumbar spine surgery* | | 6 (5) | 4 (3) | 7 (6) | 6 (5) | 4 (4) | 27 (5) |

*Only for participants who responded 'yes' to current or previous history of low back pain.
†Health professionals included nine nurses, four doctors (one GP), 3 healthcare assistants, two pharmacists, an occupational therapist, pharmacy technician, physical therapist, physician assistant, prescription manager, psychologist, respiratory therapist and a 'therapist'.
BBQ, Back Beliefs Questionnaire; GP, general practitioner; LBP, low back pain.

terms (figure 1). Overall, 41 (6%) participants provided neutral answers for all comprehension questions (ie, neither agree nor disagree).

The proportion who reported understanding the meaning of terms ranged from 35% (n=235) for the term 'disc degeneration' to 10.5% (n=71) for 'Modic changes' (figure 1A). The same proportion (88%) somewhat or completely agreed they would be worried about a serious problem with their back for at least one term and only 3% (n=17) disagreed with this statement for all terms (figure 1B). Three terms, 'disc extrusion', 'disc degeneration' and 'facet joint degeneration', elucidated the most worry about seriousness (71% of participants). Worry regarding seriousness varied only a small amount between terms, with the number of participants disagreeing with the statement varying between only 7% and 11% for all terms.

For all terms, over half the participants were worried that back pain would persist if the imaging finding was present (figure 1C). The least number of participants (51%) reported worry regarding persistence for the term 'Modic changes', while 'disc degeneration' was the term which caused worry in the greatest percent of participants

(71%). In total 84% of participants agreed they would be worried regarding persistence for at least one term, and 4% of participants disagreed with the statement for all terms.

For many imaging findings, participants also frequently reported they would be afraid to move their back. Seventy-four per cent agreed with this statement for at least one term and only 8% disagreed with this statement for all terms (figure 1D). The greatest number of participants agreed with this statement for the term 'disc bulge' (57%), and 56% of participants also agreed with this statement for the terms 'disc degeneration', 'disc protrusion', 'nerve root contact' and 'disc extrusion'. 'Modic changes' and 'mild canal stenosis' were the terms associated with least worry regarding movement (42% agreed with the statement).

## Associations between self-reported understanding of terms and perceived seriousness, risk of pain persistence and fear of movement

There was moderate correlation between understanding of terms and worry about seriousness, pain persistence and fear of movement (Spearman's r 0.48, 0.39 and 0.38,

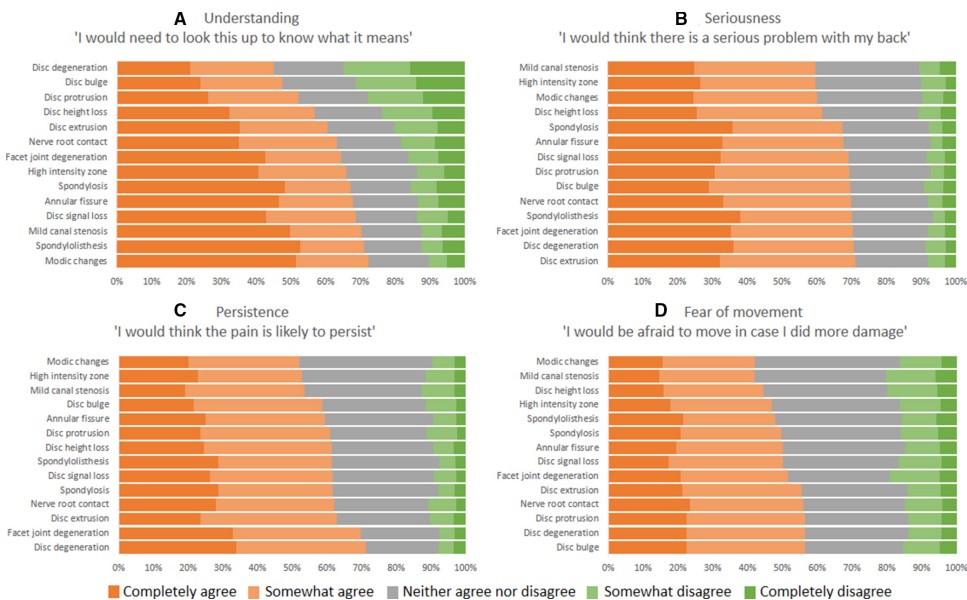

**Figure 1** Responses to questions regarding level of understanding, seriousness, persistence of pain and fear of movement for the 14 radiology terms note: for each term, we asked four questions with a common opening ('If this term was in my report I would…') to evaluate (A) understanding ('….need to look this term up to know what it means'); (B) worry about seriousness ('be worried there is a serious problem with my back'); (C) worry about persistence ('…be worried my pain would persist') and (D) fear of movement ('…be afraid to move in case I did more damage').

respectively, all p<0.001) (table 2). Strong associations were present between the responses regarding worry about seriousness and pain persistence (Spearman's r 0.63, p<0.001) and pain persistence and fear of movement (Spearman's r 0.51, p<0.001) and there was a moderate association between the responses regarding worry about seriousness and fear of movement (Spearman's r 0.42, p<0.001).

### Associations with demographic features, history of back pain and back beliefs

Results of the regression analysis are presented in table 3. Higher education level was associated with greater self-reported understanding of terms, and less concern about seriousness, pain persistence and being fearful of movement. Participants with current LBP had better understanding and less fear of movement compared with those without a history of LBP. More positive back beliefs (higher BBQ score) were associated with greater self-reported understanding of terms and less worry regarding seriousness, persistence and movement. Older

age was associated with greater worry about seriousness and pain persistence.

### Sensitivity analysis

The sensitivity analyses are shown in online supplemental tables 1–4. There was still an association between understanding of terms and worry about seriousness, pain persistence and fear of movement when we combined neutral responses with somewhat and strongly disagree with the four statements, but the correlation was weak rather than moderate (Spearman's r 0.29, 0.21 and 0.19 respectively, all p<0.001) (online supplemental table 1). Strong associations remained between worry about seriousness and pain persistence (Spearman's r 0.72, p<0.001) and pain persistence and fear of movement (0.60, p<0.001), and there was also a strong association between worry about seriousness and fear of movement (Spearman's r 0.53, p<0.001).

Combining neutral responses with somewhat disagree and strongly disagree found that higher education level was again associated with greater self-reported understanding

**Table 2** Association between the self-reported understanding of the 14 terms overall and perceived seriousness, pain persistence and fear of movement, Spearman's r correlation*

|  | Understanding | Seriousness | Pain persistence | Fear of movement |
|---|---|---|---|---|
| Understanding | 1 |  |  |  |
| Seriousness | 0.48 | 1 |  |  |
| Persistence | 0.39 | 0.63 | 1 |  |
| Fear of movement | 0.38 | 0.42 | 0.51 | 1 |

Values between 0.10 and 0.29 represent a weak association, 0.30–0.49 moderate and 0.5 and above a strong association.
*All p<0.001.

**Table 3** Association between self-reported understanding of the 14 terms, perceived seriousness, pain persistence and fear of movement and demographic details, history of back pain and back beliefs

| | Understanding | Seriousness | Pain persistence | Fear of movement |
|---|---|---|---|---|
| | Regression coefficient (95% CI) | | | |
| Education (REF-high school) | | | | |
| Trade/diploma | 0.36 (−0.01 to 0.73) | −0.23 (−0.40 to 0.86) | **−0.70 (−1.28 to 0.12)*** | −0.44 (−0.94 to 0.07) |
| Bachelor | **0.40 (0.07 to 0.72)*** | −0.31 (−0.19 to 0.81) | −0.37 (−0.85 to 0.11) | **−0.50 (−0.92 to −0.07)*** |
| Postgraduate qualifications | **0.66 (0.23 to 1.08)**** | **−0.76 (−1.38 to −0.14)*** | **−1.23 (−1.82 to −0.65)***** | **−0.94 (−1.46 to −0.42)***** |
| LBP History (REF=never) | | | | |
| Previous LBP | 0.27 (−0.21 to 0.75) | 0.31 (−0.34 to 0.96) | 0.07 (−0.56 to 0.69) | −0.30 (−0.84 to 0.25) |
| Current LBP | **0.50 (0.02 to 0.97)*** | 0.10 (−0.55 to 0.75) | −0.15 (−0.79 to 0.48) | **−0.59 (−1.14 to −0.04)*** |
| BBQ score | **0.02 (0.0003 to 0.04)*** | **−0.04 (−0.07 to −0.01)*** | **−0.08 (−0.11 to −0.04)***** | **−0.07 (−0.10 to −0.05)***** |
| Age, years | 0.002 (−0.007 to 0.01) | **0.02 (0.01 to 0.04)**** | **0.03 (0.02 to 0.04)***** | −0.004 (−0.01 to 0.01) |
| Gender (REF=female) | | | | |
| Male | 0.07 (−0.20 to 0.33) | −0.40 (−0.84 to 0.04) | −0.23 (−0.63 to 0.18) | −0.17 (−0.52 to 0.17) |

Regression estimates were obtained using the generalised linear model with binomial distribution, log link and robust estimates. Bold text indicates statistically significant results. Results rounded to two decimal points or first meaningful decimal point.
Continuous variables (BBQ score and age): For understanding, a positive regression coefficient indicates greater self-reported understanding is associated with better back beliefs (higher BBQ score) and increased age. For seriousness, pain persistence and fear of movement, a positive regression coefficient indicates greater worry/concern is associated with poorer back beliefs (lower BBQ score) and increased age.
Categorical variables: For understanding, a positive regression coefficient indicates increased self-reported understanding compared with the reference category. For seriousness, pain persistence and fear of movement, a positive regression coefficient indicates greater worry/concern compared with the reference variable.
*P<0.05, **p<0.01, ***p<0.001.
BBQ, Back Beliefs Questionnaire; LBP, Low Back Pain; REF, Reference category.

of terms but the relationship between education and seriousness, persistence and fear of movement, and the relationship of age to seriousness and persistence was no longer significant, suggesting these results should be interpreted with caution (online supplemental table 2). However, more positive back beliefs (higher BBQ score) were again associated with greater self-reported understanding of terms and less worry regarding seriousness, pain persistence and movement. In contrast to the primary analysis, the relationship between male gender became significant for seriousness and pain persistence, suggesting the relationship between gender and these questions may also be complex.

Removal of the neutral responses from the analysis altogether led to results that were in close keeping with a priori primary analysis. The only differences were that the association between understanding and seriousness and between seriousness and fear of movement became strong rather than moderate (Spearman's r 0.60 and 0.53, respectively, both p<0.001), the association between increasing age became significant for both seriousness and pain persistence, and male gender became significant for seriousness (online supplemental tables 3 and 4).

### Participants views about imaging reports
The majority of participants (n=605, 89%) agreed that people with LBP should have access to their radiology report, that it should be written in a way to facilitate lay understanding (n=586, 87%) and it would be useful to include information about the prevalence of common imaging findings in people without back pain (n=587, 87%).

### DISCUSSION
Our study found evidence of poor self-reported understanding of terms commonly used to describe usually benign findings reported in lumbar spine imaging reports. Most participants had concerns about their seriousness and associated the terms with persistence of pain and/or need to avoid (or be fearful of) movement. Higher education, more positive back beliefs and current LBP were all associated with greater self-reported understanding of the terms, while people with more positive back beliefs were also less concerned about seriousness, pain persistence and being fearful of movement. These results were robust to either combining neutral responses with disagreement with the statements or removing them from the analyses altogether.

Our findings, consistent with other literature that has found that language matters,[15 16] provide support for changing how these findings are reported. Recent online scenario-based randomised experiments that provided imaging reports for 'virtual patients' with LBP found that altering the language and/or providing epidemiological

information improved perceptions regarding LBP compared with provision of a standard report.[27 28] This now requires confirmation in randomised trials of real patients.

The radiology report may play a role in mediating the relationship between imaging and overtreatment in LBP,[7 29] and providing contextual information has been reported to reduce unnecessary repeat imaging, opioid prescription and specialist referral.[30 31] Trials that have assessed the provision of epidemiologic data or additional educational messaging,[32–36] including for LBP,[20 36 37] have been designed to improve referrer knowledge and behaviour, but none have assessed patient knowledge or other outcomes. Our study provides evidence that commonly identified and reported terms in radiology reports are also associated with misplaced patient concerns about their implications, which may also influence their healthcare decision making.

Those with current LBP had poorer back beliefs compared with those without a history of LBP or past LBP. This is consistent with previous studies that have found greater pessimism regarding recovery among those with current symptoms.[26 38–41] This may also explain their increased expectation and wish for further tests and treatment.[42]

## Strengths and limitations

To our knowledge, this is the first study to measure societal self-reported understanding of commonly used terms in lumbar spine imaging reports and their relationship to other factors such as perceived seriousness, persistence of pain and fear of movement. Although we were unable to calculate a response rate, it is likely our results are generalisable to the five English-speaking countries we included as we used age and gender quotas to reflect the make-up of each country. Additionally, the proportion of participants with a life-time history of LBP is consistent with other population-based studies,[38 40] although the prevalence of current LBP was slightly higher which may indicate greater interest in completing a survey about LBP. BBQ scores of our participants were also similar to other population-based cohorts.[38 40 41 43 44]

On the other hand, we used a survey sampling company, which may have introduced participation bias as participants receive rewards although very small ones. In addition, people who are online and familiar with computers and online questionnaires may differ from the general population. Our sample included a small number of health professionals with diverse roles and we therefore think that excluding them from our analysis would have been unlikely to change our results. We also did not consider that any meaningful conclusions could be drawn from analysing this group separately. Our study was appropriately powered and randomisation of the terms for each participant minimised any order effect bias. Where possible we adapted our survey items from other tools with established validity and reliability and the survey was piloted prior to use.

Survey measurement of public understanding of medical terms is challenging. Asking respondents whether they would need to look the term up to know what it means could be considered an even more indirect measure of self-reported understanding. As well, self-reported understanding of a medical term may not be a true reflection of participants' true understanding of terms and our study may have over or underestimated true understanding. Similarly, we did not include a measure of health literacy in our survey and we, therefore, do not know how differing health literacy skills may have influenced our findings. A large proportion of answers relating to the items were neutral (ie, neither agree nor disagree), however, only 19 (2.8%) of participants provided neutral answers for all items and they had a wide range of BBQ scores ranging from 15 to 45 suggesting that their neutral answers were likely genuine.

We also performed post hoc sensitivity analyses to determine whether our dichotomisation was an appropriate method of handling the data, and in particular, the neutral responses. The association between greater self-reported understanding of the terms in people with higher education, more positive back beliefs and current LBP, as well as the lesser concern about seriousness, pain persistence and being fearful of movement among people with more positive back beliefs indicates our findings were generally robust to altering how we handled neutral responses. The sensitivity analysis did, however, reduce the significance of some findings, notably the relationship between education and concern about terms, indicating that some of the results of the negative binomial regression analysis should be interpreted with caution.

All of the imaging findings we assessed frequently exist in the asymptomatic population, and therefore, may or may not have relevance in an individual case. However, some findings such as disc bulge, disc extrusion, Modic changes, disc protrusion and disc degeneration, occur more commonly in symptomatic populations.[19] Participants' past experiences with LBP and/or imaging of the lumbar spine and these terms may have influenced their responses.

## Implications for practice

All terms we investigated elicited concerns that were likely unwarranted. Combined with our finding that most participants would like access to their reports, more consideration is needed about how to report findings in lumbar imaging reports in people with LBP. Our finding that respondents wanted reports to be understandable to the lay person is also consistent with the results of other studies.[45 46]

Although clinicians may have reservations about allowing patients direct access to their medical information,[47] there is already increased availability and use of patient portals[48] and in many health systems this is standard practice.[49] In organisations where this has occurred, radiology reports are one of the most common items reviewed by patients accessing their information.[50] A recent addition to patient portals has been a lay-language

glossary,[51] however, with a current focus on defining anatomical structures it may not reduce patient concern regarding benign findings. Any changes to patient interpretation of radiology reports as a result of using the glossary have not been evaluated.

Moreover, there is clear evidence that the LBP beliefs and attitudes of healthcare professionals have an impact on patient beliefs and affect their clinical care.[52] A recent study found that the way lumbar spine imaging is reported can influence the beliefs of clinicians regarding the severity of the condition and their approach to management, including their views on whether surgery may be required.[53] Along with more in depth exploration of the impact of patient attitudes and beliefs regarding radiology findings, identifying and addressing the clinician factors that trigger low value care based on imaging findings are important areas for future research.

It is likely that our results may be generalisable to other regional musculoskeletal conditions where imaging also often reveals age-related changes that are common in asymptomatic people. These changes include disc bulges in the cervical spine,[54] meniscal tears in the knee[55] and supraspinatus tendinosis in the shoulder.[56] Our study could be replicated for these conditions to determine if our study findings are also applicable to them.

There is now both imperative and empirical data indicating the importance of accurately and clearly portraying the significance of imaging findings in terms that are understandable to both clinicians and patients.

## CONCLUSION

Common and usually non-serious terms in lumbar spine imaging reports are poorly understood by the general population and may contribute to the burden of LBP. Incorporating clear explanations about the implications of these findings may reduce unwarranted anxiety and reduce low-value care.

**Author affiliations**
[1]Department of Epidemiology and Preventive Medicine, School of Public Health and Preventive Medicine, Monash University, Melbourne, Victoria, Australia
[2]Monash-Cabrini Department of Musculoskeletal Health and Clinical Epidemiology, Cabrini Health, Malvern, Victoria, Australia
[3]Centre for Statistics in Medicine, Rehabilitation Research in Oxford, Nuffield Department of Orthopaedics Rheumatology and Musculoskeletal Sciences (NDORMS), University of Oxford, Oxford, UK
[4]School of Medicine and Public Health, The University of Newcastle, Newcastle, New South Wales, Australia
[5]Sydney Health Literacy Lab, School of Public Health, The University of Sydney, Sydney, New South Wales, Australia
[6]School of Public Health, The University of Sydney, Sydney, New South Wales, Australia
[7]The University of Sydney Institute for Musculoskeletal Health, Sydney, New South Wales, Australia
[8]AECC University College, Dorset, UK
[9]Centre for Pain IMPACT, Neuroscience Research Australia, Randwick, New South Wales, Australia
[10]Prince of Wales Clinical School, University of New South Wales, Sydney, New South Wales, Australia
[11]University of South Wales Faculty of Life Sciences and Education, Treforest, UK
[12]Departments of Radiology, Neurological Surgery and Health Services, School of Medicine, University of Washington, Seattle, Washington, USA
[13]UW Clinical Learning, Evidence And Research (CLEAR) Center for Musculoskeletal Disorders, University of Washington, Seattle, Washington, USA

**Acknowledgements** We acknowledge and thank Alex Gorelik for her assistance with data analysis for this study.

**Contributors** CF contributed to the conception and design of this study, the acquisition, analysis and interpretation of the data, drafting and revising the work. DAO'C contributed to the design of this study, the acquisition, analysis and interpretation of the data, drafting and revising the work. HL contributed to the conception and design of this study, the analysis and interpretation of the data and revising the work critically for content. KM contributed to the conception and design of this study, the interpretation of the data and revising the work critically for content. CM contributed to the conception and design of this study, the analysis and interpretation of the data and revising the work critically for content. DN contributed to the conception and design of this study, the interpretation of the data and revising the work critically for content. AC contributed to the conception and design of this study, the interpretation of the data and revising the work critically for content. DB contributed to the conception and design of this study, the interpretation of the data and revising the work critically for content. JJ contributed to the conception and design of this study, the interpretation of the data and revising the work critically for content. RB contributed to the conception and design of this study, the acquisition, analysis and interpretation of the data, drafting and revising the work. All authors approved the final version and agree to be accountable for all aspects of this study.

**Funding** DAO'C is supported by an Australian National Health and Medical Research Council (NHMRC) Translating Research into Practice Fellowship (APP1168749). HL is funded by an NHMRC (grant no. APP1126767); receives project funding from the Berkeley Initiative for Transparency in the Social Sciences, a programme of the Centre for Effective Global Action (CEGA), with support from the Laura and John Arnold Foundation. He is also supported by funding from the National Institute for Health Research (NIHR) Collaboration for Leadership in Applied Health Research and Care Oxford at Oxford Health NHS Foundation Trust. JJ is supported by funding from the US National Institute of Health/NIAMS (P30AR072572). CM is supported by an NHMRC Principal Research Fellowship (APP1103022). RB is supported by an NHMRC Senior Principal Research Fellowship (APP1082138).

**Competing interests** CM and RB have received grants from numerous government and not-for-profit agencies; their expenses have been covered by professional associations hosting conferences where they were speakers, and they were both investigators for a trial that received heat wraps from Flexeze at no cost.

**Patient consent for publication** Not required.

**Ethics approval** This project was approved by the Monash University Human Research Ethics Committee (ID 13896). Completion of the survey was considered consent to participate.

**Provenance and peer review** Not commissioned; externally peer reviewed.

**Data availability statement** Data are available upon reasonable request.

**ORCID iDs**
Caitlin Farmer http://orcid.org/0000-0001-6886-7784
Denise A O'Connor http://orcid.org/0000-0002-6836-122X

Hopin Lee http://orcid.org/0000-0001-5692-0314
Kirsten McCaffery http://orcid.org/0000-0003-2696-5006
Christopher Maher http://orcid.org/0000-0002-1628-7857
Dave Newell http://orcid.org/0000-0003-1462-3586
Aidan Cashin http://orcid.org/0000-0003-4190-7912
David Byfield http://orcid.org/0000-0003-0898-5337
Jeffrey Jarvik http://orcid.org/0000-0002-0528-3455
Rachelle Buchbinder http://orcid.org/0000-0002-0597-0933

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
