## [Reviewer comments · BMJ Open]

ARTICLE DETAILS

TITLE (PROVISIONAL)	Consumer understanding of terms used in imaging reports requested for low back pain: a cross-sectional survey
AUTHORS	Farmer, Caitlin; O'Connor, Denise; Lee, Hopin; McCaffery, Kirsten; Maher, Christopher; Newell, Dave; Cashin, Aidan; Byfield, David; Jarvik, Jeffrey; Buchbinder, Rachelle

VERSION 1 – REVIEW

REVIEWER	Pentz, Rebecca Emory University
REVIEW RETURNED	06-Mar-2021

GENERAL COMMENTS	This is an interesting and well written study. The survey development is well explained and excellent. Given the evidence you present in the introduction, that spinal imaging should not regularly be used, I suggest you discuss this problem in the discussion. If overuse is the real issue, misunderstanding may be a secondary problem. Also, it would be extremely valuable if you provided the results from your provider subgroup, even if small. Were providers overconcerned about non problematic findings? This would be important to note. You note in your methods section that the neutral statement 'neither agree or not agree' was grouped with the positive agreement statements. Though you single out the neutral statement in Figure 1 and in your results, I am concerned about its incorporation in any of your results. I would point out in which results you include neutral as a possible Neutral is often a response when the participant doesn't want to think carefully about the question. One small grammatical issue: p. 5 first sentence is missing a word "This is the first study to directly measure societal self-reported understanding of 14 commonly used terms found in lumbar spine imaging reports and its with worry regarding the seriousness and persistence of low back pain and being fearful of movement." Overall this is an important contribution to the research on lay understanding of medical jargon. I am not able to assess the statistical analyses and suggest a statistician review this manuscript
---

REVIEWER	Zhang, Zhan Pace University
REVIEW RETURNED	11-Mar-2021

GENERAL COMMENTS	This is a well-designed study that assessed healthcare consumers' understanding of radiology terms in low back pain (LBP). Participants from five English-speaking countries were recruited to participate in a survey. The paper is well written and easy to follow. This paper has many strengths, such as valid instruments, and an interesting and timely topic. I do, however, have several questions and comments for the authors to consider: How were the 14 radiology terms selected? Regarding participant views regarding terms, I did not see the report of difference between participants with current LBP and those without a history of LBP. This analysis would be interesting. Health literacy has effects on lay individuals' understanding of medical terms. This study, however, did not take into account this important factor. Can the authors explain why? Implications: This paper has very limited discussions on the study implications. What can medical professionals and informatics researchers do to help lay individuals interpret radiology reports? Also, given the wide adoption of the patient portal, are there any system design implications to support the comprehension of radiology terms (e.g., explaining professional terms with lay language)? This study focused on LBP specifically. Are the results generalizable to other domains? There is no discussion on the study's generalizability.
---

VERSION 1 – AUTHOR RESPONSE

Reviewer Reports

Reviewer: 1

Dr. Rebecca Pentz, Emory University

Comments to the Author:

This is an interesting and well written study. The survey development is well explained and excellent. Given the evidence you present in the introduction, that spinal imaging should not regularly be used, I suggest you discuss this problem in the discussion. If overuse is the real issue, misunderstanding may be a secondary problem.

Author's response:

We have further expanded the discussion as requested to include the impact of clinician beliefs and attitudes on LBP care and the potential triggers of overtreatment (page 19-20). We consider that misunderstanding and lack of evidence-based knowledge of the significance of abnormalities detected on imaging are significant drivers of overuse.

Reviewer comments:

Also, it would be extremely valuable if you provided the results from your provider subgroup, even if small. Were providers overconcerned about non problematic findings? This would be important to note.

Author's response

The main reason we asked this question was to ensure that the study included the general population and was not skewed to people who had provider experience in the area.

In total, there were only 27 (4%) participants included who stated they were a health care provider (9 nurses, 4 doctors including one GP, 3 nurses doctors health care assistants, 2 pharmacists, 1 occupational therapist, 1 pharmacy technician, 1 physical therapist, 1 physical therapy assistant, 1 physician assistant, 1 prescription manager, 1 psychologist, 1 respiratory therapist and 1 whose answer was 'therapy'). These data have been added as a footnote to Table 1.

Given the small number of health professionals in the study and the diversity in roles and health systems we did not consider that any meaningful conclusions could be drawn from subgroup analysis of analysing this group separately. We have now clarified this in the text.

We agree that further research investigating clinicians' views regarding these terms is needed although existing studies already suggest that clinicians may be overconcerned about non problematic findings. We have added text regarding this issue in the discussion to reflect this (page 20).

Reviewer comments:

You note in your methods section that the neutral statement 'neither agree or not agree' was grouped with the positive agreement statements. Though you single out the neutral statement in Figure 1 and in your results, I am concerned about its incorporation in any of your results. I would point out in which results you include neutral as a possible. Neutral is often a response when the participant doesn't want to think carefully about the question.

Author's response:

We considered how to dichotomise the responses at length and have clarified our thinking in the text. For self-reported understanding, we combined a neutral response (neither agree nor disagree) with somewhat or completely agree as we considered this response more likely indicated that a respondent didn't understand a term. For items regarding worry about seriousness, persistence of pain and fear of movement, we similarly combined a neutral response with somewhat or completely agree with these statements as we considered a neutral response more likely indicated that a respondent was worried about seriousness, pain persistence or was fearful of movement (Pages 9-10).

With regard to lack of care in completing the survey, we excluded any participant who provided the same answer for every item of the Back Beliefs questionnaire (BBQ). This type of response is nonsensical and likely indicates a lack of care in completing the questionnaire. This has now been clarified in the text. Furthermore, only 19 (2.8%) of the included participants answered neutrally to each question about the terms. These participants had a wide range of BBQ scores, ranging from 15-45 suggesting that, overall, they did answer the questions in the survey with care. We have added these details to the text (page 19).

Reviewer's comment:

One small grammatical issue: p. 5 first sentence is missing a word

"This is the first study to directly measure societal self-reported understanding of 14 commonly used terms found in lumbar spine imaging reports and its with worry regarding the seriousness and persistence of low back pain and being fearful of movement."

Author's response: Thank you, we have amended this (see page 4).

Overall this is an important contribution to the research on lay understanding of medical jargon. I am not able to assess the statistical analyses and suggest a statistician review this manuscript

Reviewer: 2

Dr. Zhan Zhang, Pace University

Comments to the Author

This is a well-designed study that assessed healthcare consumers' understanding of radiology terms in low back pain (LBP). Participants from five English-speaking countries were recruited to participate in a survey.

The paper is well written and easy to follow. This paper has many strengths, such as valid instruments, and an interesting and timely topic.

I do, however, have several questions and comments for the authors to consider:

Reviewer's comments:

How were the 14 radiology terms selected?

Author's response:

We reported why these were chosen on the bottom of page 7 but have clarified the wording to make this clearer, i.e., 'these terms were chosen on the basis that they are commonly found in lumbar imaging reports of people with LBP but are also common findings in asymptomatic populations and are usually considered benign or of unclear clinical significance'.

Reviewer's comments:

Regarding participant views regarding terms, I did not see the report of difference between participants with current LBP and those without a history of LBP. This analysis would be interesting.

Author's response:

This is reported in table 3 (page 15). We found that people with current LBP were more likely to report increased self-reported understanding compared to those who had never had LBP. They were also more likely to report fear of movement than those without a history of LBP. None of the other comparisons (for seriousness and persistence) between these groups were statistically significant.

Reviewer comments:

Health literacy has effects on lay individuals' understanding of medical terms. This study, however, did not take into account this important factor. Can the authors explain why?

Author's comments:

Investigating the effects of health literacy on lay individuals' understanding of medical terms was not an aim of our study. However, we agree that health literacy is an important component of understanding of medical terms, and this is likely to be a fruitful area for further research. One of the authors co-developed the Health Literacy Questionnaire (HLQ), a multidimensional tool with nine independent scales and comprising 44 items. While it would have been worthwhile adding this tool to this study we were concerned about the length of the survey and the potential reduction in completion rate if it had been included. We have added a sentence about health literacy to the discussion (page 19).

Reviewer's comments:

Implications: This paper has very limited discussions on the study implications. What can medical professionals and informatics researchers do to help lay individuals interpret radiology reports? Also, given the wide adoption of the patient portal, are there any system design implications to support the comprehension of radiology terms (e.g., explaining professional terms with lay language)?

Author's response:

Thank you for these suggested additions. We have expanded our discussion of the study implications as requested (see page 20).

Reviewer's comments:

This study focused on LBP specifically. Are the results generalizable to other domains? There is no discussion on the study's generalizability.

Authors response:

We consider that it is likely that our results could be generalizable to other musculoskeletal conditions although this requires empiric testing. We have expanded the discussion to include this point (see page 20-21).

VERSION 2 – REVIEW

REVIEWER	Pentz, Rebecca Emory University
REVIEW RETURNED	29-May-2021

GENERAL COMMENTS	I would like to thank the authors for their careful attention to the reviewers' comments. I have one remaining concern. I agree with the authors that if a participant gives the neutral answer to the question about whether they need to look up a term, that that neutral answer should be grouped with agree. However, I do not think that if a participant provides the neutral answer to a question about worry, that it should be dichotomized with agree answers. If I am asked if I am worried about something and I shrug my shoulders and answer neutral, I think this indicates a lack of worry. When I am worried I definitely state that. I recommend that an expert qualitative analyst review this method of dichotomizing answers, and if he/she agrees with me, a major revision is required to redo that analyses of the worry questions. I defer to a qualitative expert on this question. All other responses are quite adequate. One small point. Please include the percent of healthcare professionals who were included in the study. You presently say there were few, but including the percent would be helpful. This is an important manuscript and should be published.
---

REVIEWER	Zhang, Zhan Pace University
REVIEW RETURNED	30-May-2021

GENERAL COMMENTS	Thanks for addressing my comments. I think the paper is ready for publication.
--

VERSION 2 – AUTHOR RESPONSE

Reviewer: 1

Dr. Rebecca Pentz, Emory University

Comments to the Author:

I would like to thank the authors for their careful attention to the reviewers' comments. I have one remaining concern. I agree with the authors that if a participant gives the neutral answer to the question about whether they need to look up a term, that that neutral answer should be grouped with agree. However, I do not think that if a participant provides the neutral answer to a question about worry, that it should be dichotomized with agree answers. If I am asked if I am worried about something and I shrug my shoulders and answer neutral, I think this indicates a lack of worry. When I am worried I definitely state that. I recommend that an expert qualitative analysis review this method of dichotomizing answers, and if he/she agrees with me, a major revision is required to redo that analyses of the worry questions. I defer to a qualitative expert on this question. All other responses are quite adequate. One small point. Please include the percent of healthcare professionals who were included in the study. You presently say there were few, but including the percent would be helpful. This is an important manuscript and should be published.

AUTHORS' RESPONSE:

Thank you for your comments. Data on the number and percent of healthcare professionals in the study by country and overall were included in Table 1 (Page 12). In total there were 27 healthcare professionals (4% of total).

Our response to the dichotomisation question is below, given it also relates to additional editors' comments.

Reviewer: 2

Dr. Zhan Zhang, Pace University, Pace University

Comments to the Author

Thanks for addressing my comments. I think the paper is ready for publication.

AUTHORS' RESPONSE:

Thank you for your comments and recommendation for publication.

VERSION 3 – REVIEW

REVIEWER	Pentz, Rebecca Emory University
REVIEW RETURNED	15-Aug-2021

GENERAL COMMENTS	Thank you for the careful attention to my concerns about dichotomizing the data and for performing the sensitivity analyses that support your conclusions, using different approaches. The supplemental files are most helpful
--